# Investigating the High-Temperature Bonding Performance of Refractory Castables with Ribbed Stainless-Steel Bars

**DOI:** 10.3390/ma17122916

**Published:** 2024-06-14

**Authors:** Linas Plioplys, Valentin Antonovič, Renata Boris, Andrius Kudžma, Viktor Gribniak

**Affiliations:** 1Laboratory of Innovative Building Structures, Vilnius Gediminas Technical University, Sauletekio Av. 11, 10223 Vilnius, Lithuania; linas.plioplys@vilniustech.lt; 2Laboratory of Composite Materials, Vilnius Gediminas Technical University, Linkmenu Str. 28, 08217 Vilnius, Lithuania; valentin.antonovic@vilniustech.lt (V.A.); renata.boris@vilniustech.lt (R.B.); andrius.kudzma@vilniustech.lt (A.K.)

**Keywords:** refractory castable, stainless steel, ribbed bars, laboratory tests, high temperature, pull-out deformation energy, cold compressive strength

## Abstract

Refractory materials containing calcium aluminate cement (CAC) are commonly used in the metallurgical and petrochemical industries due to their exceptional mechanical resistance, even at temperatures exceeding 1000 °C, and do not require additional reinforcement. This study seeks to advance this practice by developing ultra-high-performance structures that offer building protection against fire and explosions. Such structures require bar reinforcement to withstand accidental tension stresses, and the bond performance becomes crucial. However, the compressive strength of these materials may not correlate with their bond resistance under high-temperature conditions. This study investigates the bond behavior of ribbed stainless austenitic steel bars in refractory materials typical for structural projects. The analysis considers three chamotte-based compositions, i.e., a conventional castable (CC) with 25 wt% CAC, a medium-cement castable (MCC) with 12 wt% CAC, a low-cement castable (LCC), and a low-cement bauxite-based castable (LCB); the LCC and LCB castables contain 7 wt% CAC. The first three refractory compositions were designed to achieve a cold compressive strength (CCS) of 100 MPa, while the LCB mix proportions were set to reach a CCS of 150 MPa. Mechanical and pull-out tests were conducted after treatment at 400 °C, 600 °C, 800 °C, and 1000 °C; reference specimens were not subjected to additional temperature treatment. This study used X-ray fluorescence (XRF), X-ray diffraction (XRD), and scanning electron microscopy (SEM) methods to capture the material alterations. The test results indicated that the bonding resistance, expressed in terms of the pull-out deformation energy, did not directly correlate with the compressive strength, supporting the research hypothesis.

## 1. Introduction

After over 40 years of development, refractory castables with calcium aluminate cement (CAC) define an innovative solution in heat-resistant materials. Remarkable compressive strength (exceeding 150 MPa), even in extreme temperatures above 1000 °C, determines the research innovation in developing ultra-high-performance concrete (UHPC) materials with CAC binders [1,2,3,4]. CAC-based castables have revolutionized the metallurgical and petrochemical industries, inspiring new materials science advancements [5,6,7]. The ability to mix refractory aggregates with CAC on-site ensures flexible technological solutions. Easy blending, convenient shaping, and high mixture stability determine the technological advantages that extend engineering capacities [8,9,10].

Conventional castables (CCs) incorporate aggregates with low Al_2_O_3_ content and CAC, with up to 40% aluminum oxide. Therefore, they are less expensive than medium- and low-CAC content castables with relatively expensive mixture components. However, the CC’s structural applicability is limited because of the loss of mechanical properties under temperatures between 800 °C and 1100 °C. This effect results from the binder’s dehydration and cement minerals’ recrystallization. Consequently, the initial strength may decrease more than twice [1,2,3,4]. Still, these castables remain suitable for specific industrial applications where resistance to high temperatures is required, and mechanical properties are not critical [1]. Antonovich et al. [11] discovered that adding SiO_2_ micro-particles to a fresh castable mixture enhances the compressive strength up to three times for reference castables without these additives. Reacting with SiO_2_ micro-particles, the CAC minerals form stratlingite minerals [12]. Şengül and Erdoğan [13] found a positive correlation between the stratlingite concentration and the cold compressive strength (CCS). The investigation by Suvorov et al. [6] supported this inference, indicating that the SiO_2_ modification improves the mechanical properties of refractory materials and enhances their thermal resistance. The micro-silica/alumina activation process also improves castables’ strength and sinterability temperature compared to the conventional alternatives [9,10,14].

On the other hand, refractory castables and ordinary Portland cement-based concretes share similarities despite their physical differences. Both materials contain inorganic oxides and serve essential structural purposes [1,15]. However, unlike Portland cement, where the types of minerals formed during hydration are unaffected by curing duration and temperature, the hydration of CAC results in the formation of various crystalline and amorphous hydrates, primarily influenced by the curing temperature [16]. The hydration of CAC can yield four principal temperature-dependent hydrates: CAH_10_ (CaO·Al_2_O_3_·10H_2_O), C_2_AH_8_ (2CaO·Al_2_O_3_·8H_2_O), AH_3_ (Al_2_O_3_·3H_2_O), and C_3_AH_6_ (3CaO·Al_2_O_3_·6H_2_O) [17]. Although the hydration products of these cements differ, both materials share common reinforcement principles. Refractory materials typically incorporate high-strength minerals for reinforcement, forming structures like a corundum CA_6_ network or using mullite macro-needles [2,18]. Corrosion is the problem for steel reinforcement [19,20], with stainless-steel fibers being a standard solution to solve the issue in refractory structures [21,22]. Barreiro et al. [22] found that refractory castables reinforced with stainless-steel fibers exhibit varying properties across temperatures ranging from 600 °C to 1000 °C, particularly in response to changes in fiber shape. The study highlighted that knurled stainless-steel fibers achieved the most favorable mechanical properties and pull-out test results. Shang et al. [23] suggest incorporating stainless-steel fibers into MgO-C refractories enhances flexural strength. This improvement occurs through physical embedding at room temperature and reaction bonding at high temperatures.

The rare and archival experimental work by Andión et al. [24] investigated the high-temperature corrosion resistance of reinforcement bars. Nonetheless, the mechanical resistance of refractory castables with bar reinforcement and the bond performance of stainless-steel bars in refractory castables, in particular, remain unexplored. This knowledge gap is critical in developing reliable numerical models and efficient building structures. In particular, refractory material components are crucial in ensuring the building’s integrity during fire and explosion incidents. Therefore, further investigation into the bond performance of steel bars in refractory castables is essential, and it can lead to the development of more effective building structures. Abolhasani et al. [25] investigated the mechanical characteristics of CAC-based castables under elevated temperatures. This work is noticeable in this context because of the identification of diverse strength and fracture energy tendencies. This diversity may even complicate the assessment of the bond performance of refractory castables used in structural combination with bar reinforcement. In particular, these incomplete and contradictory results ensure the hypothesis that the compressive strength of these materials may not necessarily correlate with their bond resistance under high-temperature conditions.

In other words, the discussed refractory components typically do not require additional structural reinforcement [1,2,3,4]. Unlike the existing engineering solutions, this study is dedicated to advancing building protection against fire and explosions by developing ultra-high-performance structures employing refractory castables. These structures must be reinforced to withstand accidental tension stresses, and the bond performance at high temperatures is crucial [26]. Nevertheless, there is limited research on refractory materials with stainless-steel bar reinforcement, which necessitates further investigation.

The previous study [7] identified the inability of typical structural reinforcement to realize the high-performance potential of the CAC-based refractories because of the limited resistance of the structural steel to high-temperature impact. The plain surface bars from stainless steel demonstrated unacceptable weakness in bonding with the refractory material after heating. Therefore, this study continues the pull-out test campaign [7], focusing on the bonding performance of austenitic stainless-steel ribbed bars in refractory castables having different mix proportions and strengths. This investigation considers four refractory materials: the CC, the medium-cement castable (MCC), the LCC, and the low-cement bauxite castable (LCB). This research aims to estimate the bonding efficiency of these materials after undergoing typical curing at 110 °C and four different treatment temperatures ranging from 400 °C to 1000 °C. The pull-out test program employs a setup developed by Chu and Kwan [27]. The pull-out deformation energy and the ratio of this energy to compressive strength assess the bonding efficiency of these castable materials. The X-ray fluorescence (XRF), X-ray diffraction (XRD), and scanning electron microscopy (SEM) methods capture the material changes under high temperatures.

## 2. Materials and Methods

The study by [7] demonstrated the ability of the ribbed austenitic stainless 304 steel bars to form a reliable bond with CAC-based refractory castables. Therefore, this study uses the same 8 mm ribbed bars and focuses on the refractory composition and strength effects on the reinforcement bond characteristics. This experimental program includes three chamotte-based (also known as fire clay-based) refractories, i.e., the CC, MCC, and LCC, designed for a 100 MPa CCS, and one bauxite-based LCB, designed for a 150 MPa CCS.

### 2.1. Refractory Materials

The refractory materials considered in this study have different mix proportions: the CC and MCC include 25 wt% and 12 wt% of CAC; the LCC and LCB have the same CAC content (7 wt%) but different aggregates. In addition to micro-silica and deflocculants, all the mixes (except for LCB) include 2.5 wt% of 0–0.02 mm milled quartz sand (QS, JSC Anykščių Kvarcas, Anykščiai, Lithuania); the MCC and LCC also include 5 wt% and 25 wt% of reactive alumina (RA); and the LCB has 21 wt% of a clinker G50 and 42 wt% of bauxite. Table 1 determines the mix proportions of all refractory materials. In this table, MS is the micro-silica; G50 is the crushed clinker filler 0–5 mm (Gorkal 50, Górka Cement, Trzebinia, Poland); FS20 and FS30 are the polycarboxylate ester-based deflocculants; NT is the deflocculant, anhydrous sodium tripolyphosphate Na_5_P_3_O_10_; and W is water. The manufacturing procedure was identical for all castables—a Hobart mixer jumbled the dry ingredients for 5 min; water was added according to the ASTM C860-15 ball-in-hand test method [28], followed by another 5 min of mixing.

All the castables (except for CC) used CAC Gorkal 70 (G70, Górka Cement, Trzebinia, Poland); the CC employed CAC Istra 40 (I40, Calucem, Mannheim, Germany). Table 2 provides the chemical composition of the cement specified by the manufacturers.

### 2.2. Steel Reinforcement

This study employs 8 mm ribbed austenitic stainless 304 steel bars containing 18% chromium and 8% nickel. The tension tests determine the mechanical characteristics of the reinforcement steel. The bar pieces were heated with castable specimens and pull-out cubes to determine the steel’s post-heating (cold) mechanical properties. Five bars were subjected to each temperature; unheated samples were also tested for reference. Each 30 cm long sample was subjected to tension using an electromechanical universal testing machine LFM 100 (Walter+Bai, Löhningen, Switzerland) with a 0.20 mm/s loading speed and a ±0.5% position measurement accuracy. The mechanical characteristics of the reinforcement steel were measured according to the ISO 15630-1 standard requirements [29]. A 3542-050M-100-ST extensometer (Epsilon Technology Corp., Jackson, WY, USA) with a 50 mm gauge length was used to measure the bars’ elongation.

### 2.3. Mechanical Tests

This study employed the pull-out test setup developed by Chu and Kwan [27], which was used in the previous study [7]. Figure 1a shows the testing scheme; Figure 1b displays the pull-out test setup. The testing apparatus used for pull-out tests was an electromechanical machine, H75KS (Tinius Olsen, Redhill, UK), with a 75 kN capacity and a position measurement accuracy of ±0.01%. The test apparatus loaded the bar, applying the tension deformation at a 2 mm/min rate. A 50 kN load cell measured the reaction with 0.5% precision. Two linear variable displacement transducers (LVDTs) measured the relative displacement of the bar with 0.02% precision, and the average value of their readings was used for further analysis. Figure 1a shows the positions of the LVDT devices. The signal processing device, Almemo 2890-9, equipped with a workstation computer, acquired the readings and processed data from all the devices, including the LVDTs and the load cell, every second. The testing process considered the reference samples dried at 110 °C and the alternative specimens additionally treated at 400 °C, 600 °C, 800 °C, and 1000 °C. The CCS of the 70 mm cube samples was obtained following the LST EN ISO 1927-6:2013 standard [30]. The testing apparatus, Alpha 3-3000 S (Form + Test Seidner & Co., GmbH, Riedlingen, Germany), conducted the compression tests.

This test program had two stages: the first part considered the CCS effect on the bonding performance of the reference samples; the second phase investigated the bonding parameters after treatment at 400 °C, 600 °C, 800 °C, and 1000 °C. The specimens’ preparation (including curing, drying, and heating) follows the LST EN ISO 1927-5:2013 standard requirements [31]. All the castables were de-molded after 72 h of curing at 20 ± 1 °C and dried for 72 h at 110 ± 5 °C using a 2.0 kW drying camera Snol 3.5 (Umega, Utena, Lithuania). After that, the reference samples had no additional temperature treatment, while the alternative samples were additionally heated for 5 h at particular temperatures using a 3.4 kW furnace, Snol 30/1100, with an electronic controller (Umega, Utena, Lithuania) at the 2.5 °C/min rate to 700 °C and the 5.0 °C/min rate to 1000 °C. The mechanical tests employed an equal number of samples (four) for the CCS and pull-out tests. Thus, this study uses 40 (20 CCS and 20 pull-out) specimens of the MCC, LCC, and CC materials and 7 (4 CCS and 3 pull-out) LCB cubes for the reasons described in Section 3.2.

### 2.4. Material Analysis

In addition to mechanical tests, this study employs X-ray fluorescence (XRF) and X-ray diffraction (XRD) techniques to explore the structure transformation processes behind the bonding mechanisms. The XRF technique identifies the chemical composition of the specimens. It stimulates the atoms of a material with high-energy X-rays, causing them to radiate fluorescent X-rays characteristic of their elemental makeup. The XRD equipment examines atomic and molecular configurations in crystalline substances. This process is similar to using detailed blueprints to perceive structural details within materials. Therefore, the selected castable samples were crushed using a sledgehammer into small pieces. The crushed samples were then milled with steel balls in a vertical lab planetary ball milling machine, DECO-PBM-AD-2L (Deco Equipment Co., Yueyang, China), for 20 min. The resulting mixture was extracted and passed through an ISO-3310.1 sieve with a 63 μm pore size. The processed material was then compressed into ∅37 × 3 mm cylinders using the hydraulic TP20 press (Herzog Maschinenfabr GmbH & Co., Osnabrück, Germany). The ZSX Primus IV wavelength-dispersive spectrometer (Rigaku, Tokyo, Japan) with a Rh target, end window, and a 4 kW X-ray tube ensures the XRF analysis.

The XRD samples were mixed with a 99.8% pure titanium oxide standard of anatase at a mass ratio of 10:1 (sample to anatase) to provide the calibration reference. Thus, the anatase peak heights were normalized to maintain consistency in the intensity of the central peak of anatase (the diffraction angle 2*θ* = 25.28°) across all diffractograms. This normalization enables a fair comparison of the compounds by considering their relative intensities. The analysis considered the intensity of the primary peaks. The samples were analyzed using a Dron-7 diffractometer (Bourevestnik, St. Petersburg, Russia) with Cu-Kα radiation with a wavelength of *λ* = 0.1541837 nm. The experimental conditions included a voltage of 30 kV, a current of 12 mA, and the 2*θ* diffraction angle scanning from 4° to 60° at intervals of every 2 s with an incremental increase of 0.02°.

The microstructure analysis involved scanning electron microscopy (SEM) using a JSM-7600F microscope (Jeol, Tokyo, Japan). It was conducted at two accelerating voltages, 4 kV and 10 kV, and secondary electrons were used for image formation. A Quorum Q150R ES device (Quorum Technologies Ltd., Lewes, UK) applied a thin layer of electrically conducting material to the specimen surfaces before the SEM examination.

## 3. Results and Discussion

As mentioned in Section 2.3, the pull-out test program incorporates two stages. The first stage considers the CCS effect on the bond performance of the unheated samples, clarifying the object for the temperature treatment. The second stage investigates the effect of temperature on the relationship between the CCS and bonding efficiency expressed in pull-out deformation energy. In addition, the experimental campaign includes a detailed analysis of the structural transformations of the refractory materials responsible for mechanical performance. The following subsections discuss each investigation stage separately.

### 3.1. Mechanical Resistance of the Bar Steel

Figure 2 shows the tensile test results of reinforcement bars after high-temperature treatments; the “Reference” diagrams represent the unheated samples. The figure indicates a minor deterioration in the mechanical performance of the steel samples until the treatment conditions reach 600 °C, which is consistent with the results obtained from previous research [7,32]. With further temperature increase, there is a significant strength degradation, with almost a 50% and 75% reduction in yield strength corresponding to the 800 °C and 1000 °C treatments. On the other hand, the stainless steel exhibits a ductile deformation response, and the ultimate strains also increase with the temperature, exceeding 0.3 value after heating to 1000 °C. Notably, the maximum residual strength of the stainless steel after the maximum temperature exceeds 600 MPa, making it a promising alternative for reinforcing structures subjected to extreme conditions. Further investigations may employ non-destructive X-ray microtomography methods, investigating microstructural changes in the reinforcement steel, e.g., [33], which are beyond the scope of this study.

### 3.2. The CCS Effect on the Pull-Out Bonding Performance without Heat Treatment

During the first analysis stage, the reference compression and pull-out samples from all castables, cured at 110 °C, are examined to determine the CCS effect on bond performance. The analysis parameters are the pull-out deformation energy, the CCS, and normalized deformation energy regarding the CCS. The following equation determines the pull-out deformation energy:(1)$$E_{u}=\int_{0}^{u_{l}}Pdu=\frac{1}{2}\sum_{i=2}^{n_{l}}\left(P_{i}+P_{i-1}\right)\left(u_{i}-u_{i-1}\right),$$

In Equation (1), *u* is the pull-out displacement, **P** is the applied load, and *u_l_* is the reference displacement. This study assumes the same boundary deformation, *u_l_* = 15 mm, for all considered cases that unify the energy analysis. The calculations employ the trapezoidal approximation rule to calculate the area under the load-displacement curve. This approximation is appropriate because of the sufficient quantity of calculation points—the assumed loading speed and monitoring frequency (Section 2.3) ensure *n_l_* = 450 measurements until the pull-out displacement reaches 15 mm.

Figure 3 represents the results of the compression and pull-out tests. The CCS test results (Figure 3a) reveal the adequacy of the mix design to achieve the target strength value. The wickers on the diagrams correspond to the standard deviation obtained from four samples. The CC, MCC, and LCC test samples showed a slight overperformance in strength. However, these results correspond to 70 mm cube compression tests, so some reduction in strength is expected for the standardized 150 mm cube samples.

The load-displacement diagrams in Figure 3b determine three essential points as follows:

The tests resulted in the pull-out failure of all samples. This outcome supports the representativeness of the testing parameters to investigate the concrete resistance mechanisms. This observation aligns with the findings of the previous study [7] and results from the relatively smooth shape of the surface of the stainless-steel ribbed bars (Section 2.2) when compared to typical construction steel reinforcement [7].All pull-out samples show comparable ultimate load-bearing capacities of the bond despite differences in compressive strength (Figure 3a).Figure 3b demonstrates a significant scatter and differences in the shape of the diagrams, which cannot determine a quantitative measure for comparison purposes. This study employs the pull-out deformation energy, i.e., the area beneath the load-deformation diagram, Equation (1), to provide a comparative analysis. The 15 mm displacement *u_l_* terminates the deformation assessment area to make the energy estimations equivalent. The vertical dashed line in Figure 3b highlights this limit.

Figure 4a shows the deformation energy assessment results. As previously considered (Figure 3a), the standard deviation determines the length of the wickers. According to the graph, the LCC samples perform better than the alternative castables designed for a 100 MPa CCS. However, despite being designed for a 150 MPa strength, the LBC specimens do not show exceptional results due to the substantial scatter represented by the overlapping wickers. Figure 4b presents the deformation energy normalization results, which help improve the comparison adequacy of the CCS. The normalization reveals the outstanding bonding efficiency of the LCC samples, which becomes statistically significant compared to the other castables. These outcomes form the following conclusions:

The pull-out test results show significantly different bond performance despite the nominally identical strengths of the CC, MCC, and LCC refractory materials (Figure 3a). This outcome suggests that the bond resistance is predominantly affected by the chemical composition of the materials rather than the CCS.Notwithstanding a substantial increase in strength (Figure 3a), the LCB samples do not demonstrate an exceptional bond performance compared to other castables. This observation strengthens the impression about the materials-based nature of the bonding resistance mechanisms. However, because of the unsatisfying bond performance, the LCB samples were excluded from further analysis of the impact of temperature on bonding resistance.

### 3.3. The Temperature Effect on the Bond Performance of Different Refractory Materials

The testing procedures are consistent with the initial testing stage (Section 3.2), and the reference results of the CC, MCC, and LCC samples remain unchanged. This investigation stage includes the addition of heat treatment procedures at four different temperatures: 400 °C, 600 °C, 800 °C, and 1000 °C. These tests were conducted after the heated specimens were cooled in the laboratory for at least one day. As shown in Figure 5a, the compression test results indicate that the LCC sample exhibited a significant increase in the CCS, with an average strength of 170 MPa after the 1000 °C treatment. In contrast, the compressive strength of the CC and MCC samples did not change significantly, with strengths varying between 90 MPa and 115 MPa. This result aligns with the previous findings [7,34]. However, the pull-out test results demonstrate a diverse trend.

Figure 5b–d show the pull-out test results of the castables subjected to the temperature treatment. Similarly, as Figure 3b reveals, quantifying these results may be only qualitative. Therefore, Figure 6a shows the pull-out deformation energy estimation results corresponding to a 15 mm displacement. The bond performance of the LCC samples significantly outperforms the counterparts considered in this study. This outcome aligns with the strength increase in Figure 5. Therefore, Figure 6b shows the normalized energy values regarding the corresponding strength values (Figure 5a) to determine the significance of the relationship between the deformation energy *E_u_* (Equation (1)) and the CCS. The remarkable transformation in the relative LCC bond resistance emerges after the specimens are subjected to 800 °C and 1000 °C treatments when the relative bond efficiency of the MCC samples overtakes the LCC results (Figure 6b). This finding determines the object of the following analysis.

### 3.4. The Castable Structure Transformation Analysis

This section utilizes XRF, XRD, and SEM techniques to identify the chemical and microstructural changes in the refractory materials treated at 800 °C and 1000 °C, explaining the bond resistance transformations in Figure 6b. Table 3 lists the chemical composition of the castables (independent of the temperature conditions) obtained by XRF.

The LCC sample has a substantial aluminum oxide (Al_2_O_3_) content of 56.9 wt% and a silica (SiO_2_) content of 32.4 wt%, indicating that it is a highly efficient refractory material. The MCC sample has a slightly lower Al_2_O_3_ content (49.8 wt%) but a higher SiO_2_ concentration (37.0 wt%), which may affect the temperature changes in its thermal resistance and mechanical strength differently from the LCC. The CC sample has the lowest Al_2_O_3_ content (41.4 wt%) and a relatively high SiO_2_ content (34.6 wt%), which reduces its refractoriness and thermal durability.

The above results suggest that the LCC composition is potentially the most favorable for high-temperature applications due to relatively high Al_2_O_3_ content. The MCC offers a viable alternative with a balanced composition. Meanwhile, due to its compositional constraints, the CC appears less suitable for the most demanding thermal environments.

Figure 7 shows the XRD analysis patterns of the CC, MCC, and LCC samples treated at 800 °C and 1000 °C. The results of XRD analysis show the presence of the following main crystalline products of the refractory castable: mullite (with the atoms’ plane distances of two primary peaks *d* = 0.342 nm and 0.339 nm); corundum (*d* = 0.255 nm and 0.160 nm); anorthite (*d* = 0.320 nm and 0.326 nm); gehlenite (*d* = 0.284 nm and 0.306 nm); quartz (*d* = 0.334 nm and 0.425 nm); and tridymite (*d* = 0.410 nm and 0.325 nm). In Figure 7, the additional anatase (*d* = 0.352 nm and 0.237 nm) was used as an internal standard (a relative norm) to estimate and compare the relative contents of the crystal components in the samples.

Table 4 and Table 5 show the relative estimation (in the anatase regard) of the content of the tested castables’ crystalline phases, considering the identified phases’ primary peak intensities. In these tables, the total amount of refractory minerals, ∑N, is the sum of anorthite, mullite, gehlenite, and corundum, and ∑Q is the sum of quartz and tridymite. These tables indicate that the mineral content changes in all castables, with temperatures increasing from 800 °C to 1000 °C. These changes reveal the effect of temperature on the refractory materials’ crystal structure and phase composition. In particular, the relative amount of corundum is the highest in the MCC sample after 1000 °C, indicating a favorable phase transformation for the mechanical performance [35]. The MCC samples also have a more significant amount of mullite and the refractory minerals, ∑N, than the CC and LCC counterparts, independent of the treating temperature. The total amount of SiO_2_ minerals ∑Q was lower in the MCC compared to the results of the CC and LCC. These observations may explain the relative increase in the bond performance of the MCC samples observed in Figure 6b. These pull-out test results may also indicate that the corundum and ∑N mineral concentrations more significantly affect the bond performance than the CCS. Still, this inference requires a more detailed experimental study.

Figure 8 shows the MCC and LCC microstructure transformation after treatments at 800 °C and 1000 °C. It also includes SEM images near the contact surface with the reinforcement bar and inside the concrete sample. The SEM analysis revealed the apparent densification of the castable microstructure near the bar surface (marked with a dashed line). Inside the castable sample, the material microstructure becomes less dense, where the distribution of crystals is more chaotic and porous (Figure 8a,c,e,f). These results are characteristic of both refractory materials. This result aligns with the previous findings [7]. Figure 8 also reveals the qualitative changes in the material microstructure around the reinforcement bar after the different temperature treatments. These changes are essential because they may cause changes in the bonding resistance (Figure 6). In particular, the comparative analysis of the microstructure transformation reflects substantial alterations in the LCC sample (Figure 8f,h); however, these changes are not so apparent in the MCC sample (Figure 8b,d). This observation aligns with the XRD results, indicating the rise in the corundum content in the MCC sample heated at 1000 °C compared to its counterpart exposed at 800 °C. Thus, the MCC samples are prone to demonstrate an increased ratio between deformation energy and the CCS compared to the LCC and CC counterparts (Figure 6b).

The literature results [35,36] support the above inference, relating the increase in strength and deformation capacity of CAC-based refractory materials with the formation of anorthite, mullite, and corundum, which provides the background for developing efficient protective refractory structures with bar reinforcement that are resilient to high temperatures and excessive mechanical loads, characteristic of fire and explosion conditions in building structures. These results also reveal the limited correlation between the pull-out deformation energy and the CCS, determining the object for further exploration.

At the same time, the castable choice for industrial applications accounts for the thermal resistance, mechanical strength, and durability properties only in extreme conditions. Under less demanding restrictions, it depends on balancing cost and performance. Thus, the prices of the castable mix components in this study, estimated in March 2024, may reveal the objective situation with the material choice. The following expenses were obtained for a ton of dry mixture: CC = EUR 507, MCC = EUR 623, LCC = EUR 733, and LCB = EUR 1040. In unheated conditions (Figure 4a), these materials may virtually ensure 0.540, 0.423, 0.538, and 0.317 deformation energy per Euro (Nm/EUR). Still, the material choice in specific cases depends on the operating temperature range.

### 3.5. The Essential Contribution of This Study to the Research Field

The typical refractory solutions do not require structural reinforcement [1,2,3,4]. Unlike the existing practice, this study is dedicated to advancing building protection against fire and explosions by developing ultra-high-performance structures employing refractory castables with bar reinforcement. However, the literature analysis has identified the lack of test results on the bond performance of stainless-steel ribbed bars in refractory castables. Thus, this study considers four refractory materials (CC, MCC, LCC, and LCB) and makes the following essential contributions to the research field:

The austenitic stainless 304 steel bars ensure a reliable bond with all the considered refractory materials and can be used as the primary reinforcement in the developed protective structures.The CCS of the CC, MCC, and LCC samples may be identical in unheated conditions. However, the pull-out tests reveal that these castables have significantly different bond performances, indicating that the mineral composition may impact bond resistance more than compressive strength.The LCC samples treated at 800 °C and 1000 °C show substantial transformation of the deformation energy and the CCS ratio, which is not characteristic of the CC and MCC specimens. Comparative SEM analysis of the LCC and MCC samples highlights significant microstructural changes in the LCC after the 1000 °C treatment. The XRD analysis identifies opposite mullite and corundum concentration tendencies in the LCC and MCC materials. Although the latter minerals may increase strength, experiments show that the mullite and corundum concentrations impact bond performance more significantly than the CCS.

The last two findings revealed the absence of the hypothesized straightforward correlation between the CCS and the pull-out deformation energy. Each considered refractory material was designed to have a specific CCS but varying amounts of CAC, which may affect their bonding performance after exposure to high temperatures.

## 4. Conclusions

Examining the bonding performance of ribbed bars made of austenitic stainless steel, this experimental study considers four types of refractory materials, namely, the conventional castable (CC), the medium-cement castable (MCC), the low-cement castable (LCC), and the low-cement bauxite castable (LCB). The obtained results led to the following main conclusions:

The austenitic stainless 304 steel bars ensure a reliable bond with all the considered refractory materials. They can be used as the primary reinforcement in building protection against fire and explosions by developing ultra-high-performance structures employing refractory castables.The hypothesized straightforward correlation between the cold compressive strength and the pull-out deformation energy does not exist, and the mineral composition may impact bond resistance more than compressive strength. This result describes the object for further research and optimization.

## Figures and Tables

**Figure 1 materials-17-02916-f001:**
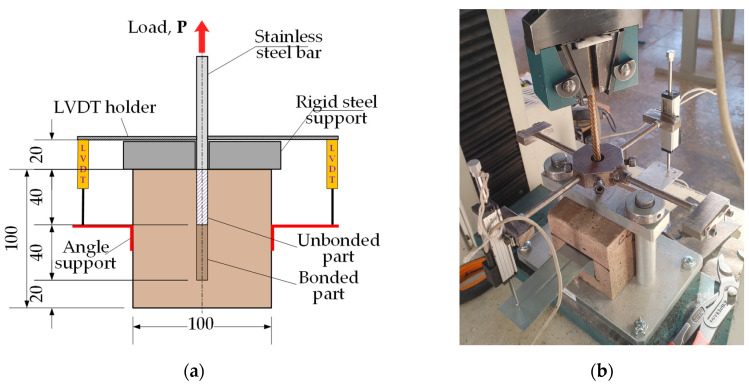
Pull-out tests: (**a**) testing scheme; (**b**) test setup.

**Figure 2 materials-17-02916-f002:**
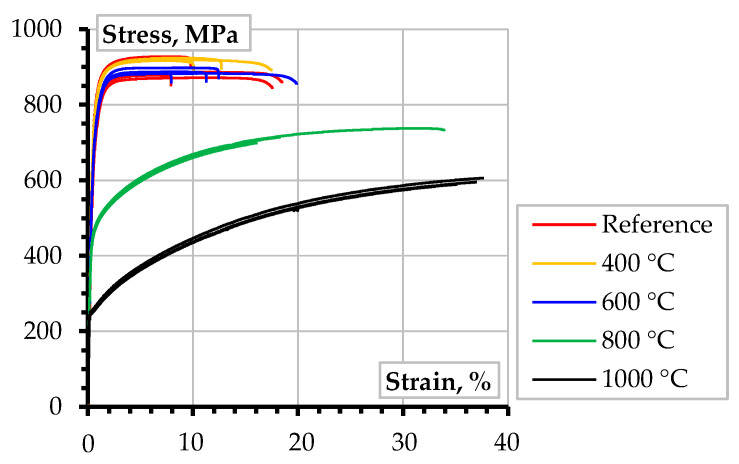
The tensile test results of 8 mm stainless 304 steel ribbed bars after temperature treatments.

**Figure 3 materials-17-02916-f003:**
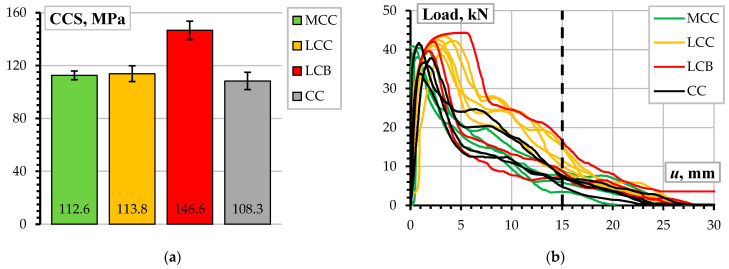
The mechanical test results of the unheated test samples: (**a**) CCS results; (**b**) load-pull-out displacement diagrams.

**Figure 4 materials-17-02916-f004:**
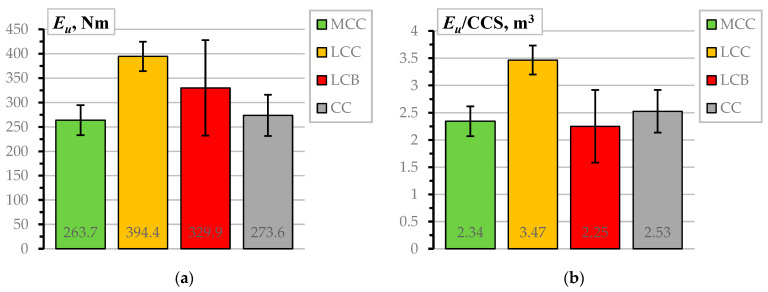
The pull-out deformation energy assessment results: (**a**) the estimated deformation energy; (**b**) the deformation energy normalized by CCS.

**Figure 5 materials-17-02916-f005:**
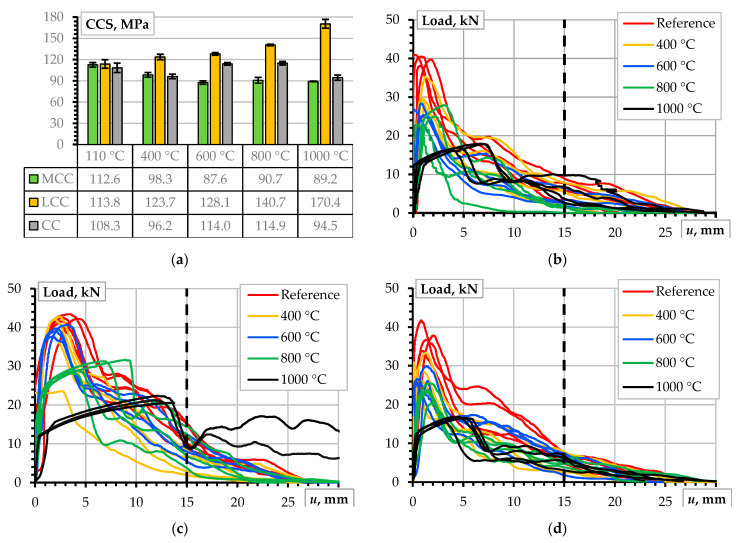
Mechanical test results of the castables subjected to temperature treatment: (**a**) the CCS of different mixture castables; (**b**–**d**) the pull-out test results of the MCC, LCC, and CC samples.

**Figure 6 materials-17-02916-f006:**
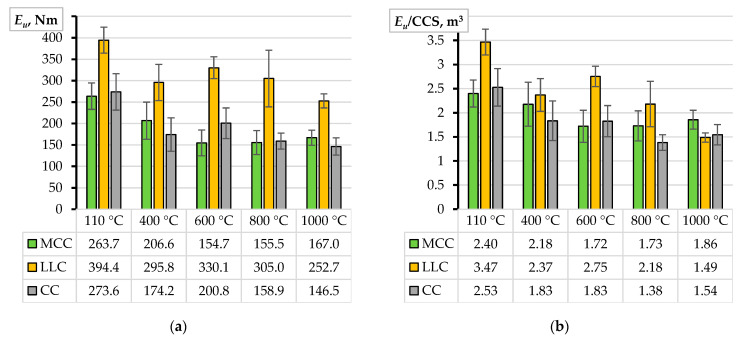
The pull-out deformation energy after temperature treatments: (**a**) the deformation energy assessment results; (**b**) the deformation energy normalized by the CCS.

**Figure 7 materials-17-02916-f007:**
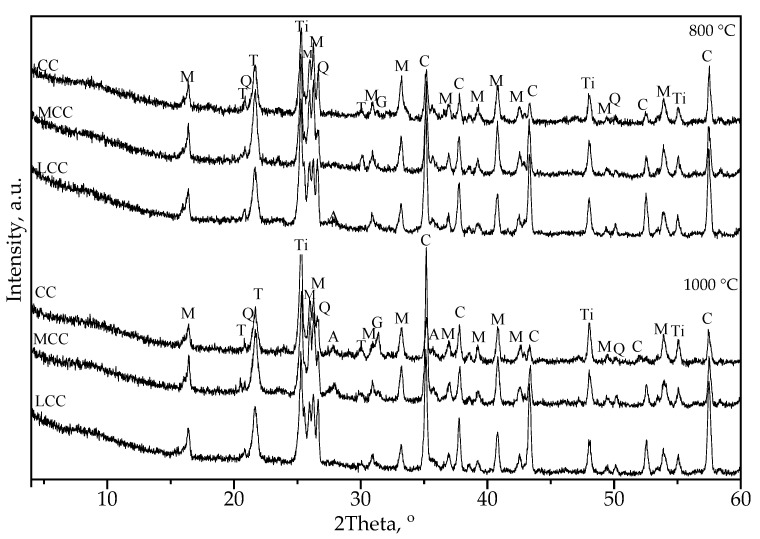
The XRD patterns of the castables after the 800 °C and 1000 °C treatments. Note: M—mullite, C—corundum, Q—quartz, G—gehlenite, A—anorthite, T—tridymite, Ti—anatase.

**Figure 8 materials-17-02916-f008:**
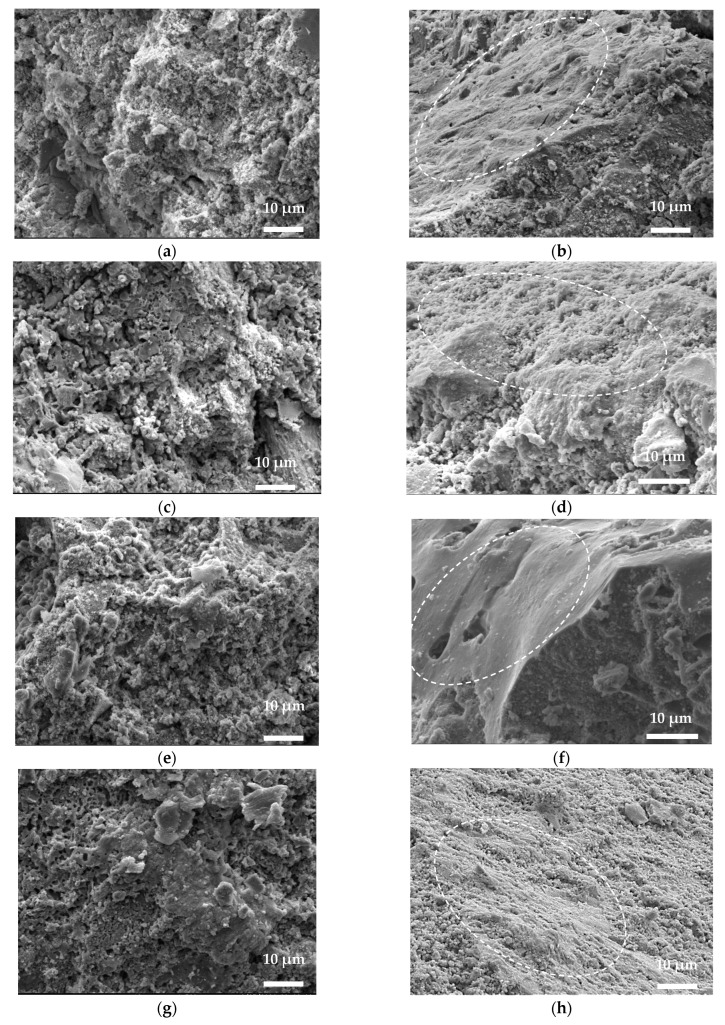
The SEM analysis results (×1500 magnification): (**a**–**d**) MCC and (**e**–**h**) LCC; (**a**,**b**,**e**,**f**) treated at 800 °C and (**c**,**d**,**g**,**h**) treated at 1000 °C; (**a**,**c**,**e**,**g**) located far away from the bar and (**b**,**d**,**f**,**h**) located near the bar.

**Table 1 materials-17-02916-t001:** Mix proportions of refractory castables (wt%).

Mix	CAC		QS	MS	RA	Fire Clay BOS 145		G50	Bauxite		FS20	FS30	NT	W
	G70	I40				0–0.14 mm	0–5 mm		0–1 mm	1–3 mm				
MCC	12	–	2.5	3	5	12	65.5	–	–	–	0.1	–	0.1	7.5
LCC	7	–	2.5	5	25	–	60.5	–	–	–	0.1	–	0.1	6.5
LCB	7	–	–	5	25	–	–	21	21	21	0.1	–	0.1	6.5
CC	–	25	2.5	2.5	–	10	60.0	–	–	–	–	0.1	–	8.0

**Table 2 materials-17-02916-t002:** CAC chemical composition (wt%).

Constituent	Gorkal 70 (G70)	Istra 40 (I40)
Al_2_O_3_	69–71	37–42
CaO	28–30	36–40
Fe_2_O_3_	<0.3	13–18
SiO_2_	<0.5	≤6
MgO	–	<1.5
SO_3_	–	<0.4
Na_2_O + K_2_O	<0.5	–

**Table 3 materials-17-02916-t003:** XRF analysis results.

Constituent	MCC	LCC	CC
Na_2_O	0.221	0.209	0.163
MgO	0.835	0.581	0.941
Al_2_O_3_	49.8	56.9	41.4
SiO_2_	37.0	32.4	34.6
K_2_O	0.699	0.645	0.612
CaO	4.35	2.73	10.8
TiO_2_	1.17	1.06	1.46
Fe_2_O_3_	1.54	1.24	5.22

**Table 4 materials-17-02916-t004:** XRD recognized minerals after 800 °C heating.

Name	Anorthite 28.02°	Mullite 26.26–30°	Gehlenite 31.38–42°	Corundum 35.15–18°	Tridymite 21.68–70°	Quartz 26.62–66°	TiO_2_ 25.32–34°	∑Q	∑N
MCC	0.000	370.0	26.60	400.0	309.0	162.0	440.0	471.0	796.6
LCC	40.00	295.0	0.000	390.0	217.0	300.5	440.0	517.5	725.0
CC	0.000	367.0	37.60	238.0	230.0	249.0	440.0	479.0	642.6

**Table 5 materials-17-02916-t005:** XRD recognized minerals after 1000 °C heating.

Name	Anorthite 28.02°	Mullite 26.26–30°	Gehlenite 31.38–42°	Corundum 35.15–18°	Tridymite 21.68–70°	Quartz 26.62–66°	TiO_2_ 25.32–34°	∑Q	∑N
MCC	59.00	361.5	37.00	634.0	255.5	181.0	440.0	436.5	1092
LCC	0.000	296.5	0.000	495.5	218.0	370.0	440.0	588.0	792.0
CC	44.00	351.0	135.0	254.0	248.0	196.0	440.0	444.0	784.0

## Data Availability

The authors will make the raw data supporting this article’s conclusions available upon request.

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
