# Peer review of "Investigating the High-Temperature Bonding Performance of Refractory Castables with Ribbed Stainless-Steel Bars"

_materials, 2024, doi:10.3390/ma17122916_

Round 1
Reviewer 1 Report
Comments and Suggestions for Authors
This study forms part of a comprehensive research initiative aimed at developing ultra-high-performance composite materials tailored to protect building structures from fire and explosions. Given the rigorous demands of this application, structural reinforcement becomes imperative, with particular emphasis placed on bond performance at elevated temperatures.Significantly, existing literature overlooks this critical issue, likely due to the specialized nature of the involved industrial processes. To address this knowledge gap, we extended our previous pull-out test program, assessing the bond behavior of ribbed stainless austenitic steel bars across various refractory materials.In this study, Linas et al. investigated three chamotte-based compositions: conventional castable (CC) containing 25% CAC, medium-cement castable (MCC) with 12 wt% CAC, and low-cement castable (LCC), alongside a low-cement bauxite-based castable (LCB), with LCC and LCB castables comprising 7 wt% CAC. The formulations of the first three refractory compositions were meticulously tailored to ensure a cold compressive strength (CCS) of 100 MPa. Meanwhile, the proportions of the LCB mix were adjusted to achieve a CCS of 150 MPa. This is a nice draft, here are my comments:
In the introduction, it's essential to cite relevant non-destructive testing methods. One such method is detailed in the study by Zhang, Zhiguo, et al., titled "Multiscale characterization of the 3D network structure of metal carbides in a Ni superalloy by synchrotron X-ray microtomography and ptychography" published in Scripta Materialia (2021).
Regarding Figure 1(b) and Figure 2(b), as they contain duplicate information, it is advisable to either delete one picture or merge the two to avoid redundancy.
The quality of the SEM image in Figure 9 needs enhancement. Consider improving the resolution or adjusting the contrast to ensure clarity and detail.
Figure 8 may not be reader-friendly. Consider merging the conditions of 800℃ and 1000℃ into a single figure for easier comparison and comprehension.
The section title "Discussion of the Results" can be revised to "Results and Discussion" for clarity and alignment with standard academic conventions.
Provide more detailed information on the heat treatment process used in the study to enhance understanding and reproducibility.
Comments on the Quality of English Language
Minor editing of English language required.
Author Response
General note: The Authors have diligently addressed all constructive comments from the eminent Reviewers. The green color highlights all text corrections, ensuring no comment has been overlooked.
Reply to Reviewer #1
Note: The authors deeply value the Reviewer’s comments, which have significantly enhanced the quality of the presentation.
Comment: This study forms part of a comprehensive research initiative to develop ultra-high-performance composite materials tailored to protect building structures from fire and explosions. Given the rigorous demands of this application, structural reinforcement becomes imperative, with particular emphasis placed on bond performance at elevated temperatures. Significantly, existing literature overlooks this critical issue, likely due to the specialized nature of the involved industrial processes. To address this knowledge gap, we extended our previous pull-out test program, assessing the bond behavior of ribbed stainless austenitic steel bars across various refractory materials. In this study, Linas et al. investigated three chamotte-based compositions: conventional castable (CC) containing 25% CAC, medium-cement castable (MCC) with 12 wt% CAC, and low-cement castable (LCC), alongside a low-cement bauxite-based castable (LCB), with LCC and LCB castables comprising 7 wt% CAC. The formulations of the first three refractory compositions were meticulously tailored to ensure a cold compressive strength (CCS) of 100 MPa. Meanwhile, the proportions of the LCB mix were adjusted to achieve a CCS of 150 MPa. This is a nice draft; here are my comments:
Comment: In the introduction, citing relevant non-destructive testing methods is essential. One such method is detailed in the study by Zhang, Zhiguo, et al., titled "Multiscale characterization of the 3D network structure of metal carbides in a Ni superalloy by synchrotron X-ray microtomography and ptychography" published in Scripta Materialia (2021).
Answer: The Authors appreciate this suggestion. The recommended reference was added to discuss further research perspectives in Section 3.1.
Comment: Regarding Figures 1(b) and 2(b), as they contain duplicate information, it is advisable to either delete one picture or merge the two to avoid redundancy.
Answer: The Authors acknowledge this thoughtful suggestion. Figure 1 was removed because of the standard setup of the tension tests. The corresponding comment was added to Section 2.2.
Comment: The quality of the SEM image in Figure 9 needs enhancement. Consider improving the resolution or adjusting the contrast to ensure clarity and detail.
Answer: The Authors appreciate this note. The resolution of the microstructure images in Figure 8 (former Figure 9) was improved; more characteristic SEM images were added, highlighting the microstructure densification zones near the bar surface.
Comment: Figure 8 may not be reader-friendly. Consider merging the conditions of 800℃ and 1000℃ into a single figure for easier comparison and comprehension.
Answer: The Authors accepted this suggestion and rearranged Figure 7 (former Figure 8) accordingly. So, all XRD lines were merged into a single diagram.
Comment: The section title "Discussion of the Results" can be revised to "Results and Discussion" for clarity and alignment with standard academic conventions.
Answer: This suggestion was accepted.
Comment: Provide more detailed information on the heat treatment process used in the study to enhance understanding and reproducibility.
Answer: The Authors sincerely appreciate this comment. The corresponding explanation was added in Section 2.3.
Comment: Minor editing of English language required.
Answer: The Authors carefully checked the text, improving its clarity and writing style to the best of their abilities. The green color highlights all modifications.
Reviewer 2 Report
Comments and Suggestions for Authors
The manuscript is a continuation/complement of previous work as cited and referenced several times in the text. It is well written and detailed, showing the properties of different materials and their potential use. The work can be accepted as is.
Author Response
General note: The Authors have diligently addressed all constructive comments from the eminent Reviewers. The green color highlights all text corrections, ensuring no comment has been overlooked.
Reply to Reviewer #2
Comment: The manuscript is a continuation/complement of previous work as cited and referenced several times in the text. It is well-written and detailed, showing different materials' properties and potential use. The work can be accepted as is.
Answer: The Authors sincerely thank the Reviewer for sharing his/her time and knowledge and positively assessing the manuscript.
Reviewer 3 Report
Comments and Suggestions for Authors
The paper's title clearly describes/reflects the research topic which otherwise has been a kind of continuation of the Authors' previous/last study (cited as Ref. 7) on testing quite many necessary and typical important properties of their prepared refractory castables with calcium aluminate cements reinforsed with ribbed stainless steel bars.
The testing methodology followed the test programme developed by Chu & Kwan (cited as Ref. 26; 2018) and used highly sophisticated techniques with a highly professional manner.
Description of the obtained results are all relevent and well grounded supporting well the conclusions.
The text is concise, well structured and written without almost any misprints or other errors. Probably the unit microns could be replaced with micrometers.
All in all, the paper quality is high with fresh scientific results and well prepared for publication in this journal.
Author Response
General note: The Authors have diligently addressed all constructive comments from the eminent Reviewers. The green color highlights all text corrections, ensuring no comment has been overlooked.
Reply to Reviewer #3
Comment: The paper's title clearly describes/reflects the research topic, which otherwise has been a kind of continuation of the Authors' previous/last study (cited as Ref. 7) on testing quite many necessary and typical important properties of their prepared refractory castables with calcium aluminate cements reinforced with ribbed stainless steel bars.
The testing methodology followed the test programme developed by Chu & Kwan (cited as Ref. 26; 2018) and used highly sophisticated techniques with a highly professional manner.
Description of the obtained results are all relevent and well grounded supporting well the conclusions.
The text is concise, well structured and written without almost any misprints or other errors.
Overall, the paper quality is high, with fresh scientific results, and well prepared for publication in this journal.
Answer: The Authors sincerely thank the Reviewer for sharing his/her time and knowledge. They appreciate the comments and suggestions, which have substantially improved the article.
Comment: The unit microns could probably be replaced with micrometers.
Answer: The Authors acknowledge this error. The unit µ was corrected to µm in Section 2.4.
Reviewer 4 Report
Comments and Suggestions for Authors
The article presents the results of tests on the bonding of stainless steel reinforcing bars with selected refractory materials.
I am not competent to evaluate the English language, but in my opinion the article is written in a language whose style is often incomprehensible to me. That's why I think it's recommended to have your language checked by a native speaker.
In general, the research part of the article does not raise any objections in my opinion. However, the content is presented more like a technical research report. There is no adequate discussion of the results in relation to the existing state of knowledge. There is no discussion showing what new research results bring to the existing state of knowledge. There is no comparison of the obtained results with the results of alternative solutions. I believe that such a discussion would significantly improve the scientific level of the article.
Abstract: The abstract is too long and contains unnecessary information. In the abstract, it is unnecessary to explain why this research was undertaken. This makes it even more unnecessary to refer to References (the DOI of the article is provided). The abstract should contain concise information about: what was researched, how it was researched and what was the effect of this research. I think that the authors should revise the abstract.
Conclusions are too long and intricately written. Conclusion 4 does not result from the presented research results at all and I believe that it should be removed.
Author Response
General note: The Authors have diligently addressed all constructive comments from the eminent Reviewers. The green color highlights all text corrections, ensuring no comment has been overlooked.
Reply to Reviewer #4
Note: The Authors sincerely thank the Reviewer for sharing his/her time and knowledge. They also profoundly value constructive comments, which have significantly enhanced the presentation quality.
Comment: The article presents the results of tests on the bonding of stainless steel reinforcing bars with selected refractory materials.
I am not competent to evaluate the English language, but in my opinion the article is written in a language whose style is often incomprehensible to me. That's why I think it's recommended to have your language checked by a native speaker.
Answer: The Authors carefully checked the text, improving its clarity and writing style to the best of their abilities. The green color highlights all modifications.
Comment: In general, the research part of the article does not raise any objections in my opinion. However, the content is presented more like a technical research report. There is no adequate discussion of the results in relation to the existing state of knowledge. There is no discussion showing what new research results bring to the existing state of knowledge. There is no comparison of the obtained results with the results of alternative solutions. I believe that such a discussion would significantly improve the scientific level of the article.
Answer: The Authors sincerely appreciate this thoughtful comment. The following essential corrections were dedicated to improving the presentation clarity:
- The Abstract was rewritten, clarifying the essential contribution of this study to engineering knowledge.
- The last three paragraphs of the Introduction were rearranged for the same purpose.
- Section 3.5 was introduced to strengthen the scientific message of this experimental work.
Comment: Abstract: The abstract is too long and contains unnecessary information. In the abstract, it is unnecessary to explain why this research was undertaken. This makes it even more unnecessary to refer to References (the DOI of the article is provided). The abstract should contain concise information about what was researched, how it was researched, and what was the effect of this research. I think that the authors should revise the abstract.
Answer: This comment is understandable, and the Authors substantially reworked the Abstract. They also deleted the unnecessary reference to the previous publication and reworked the description to reflect the essential contribution of the obtained results to the current state of knowledge. Unfortunately, this modification did not shorten the Abstract but made it clearer.
Comment: The conclusions are too long and intricately written. Conclusion 4 does not result from the presented research results at all, and I believe that it should be removed.
Answer: The Authors accepted this valuable comment. They introduced Section 3.5 to strengthen the scientific message of this experimental work and rearranged Conclusions to highlight the essential novelty of the obtained experimental results.
Round 2
Reviewer 1 Report
Comments and Suggestions for Authors
The manuscript has been greatly improved, I have no further comments.
Author Response
Reply to Reviewer #1
Comment: The manuscript has been greatly improved; I have no further comments.
Answer: The Authors sincerely appreciate the favorable evaluation of this work.
Reviewer 4 Report
Comments and Suggestions for Authors
The authors provided a revised article and responded to my comments.
Below is my comment on the authors' answers:
The abstract has been corrected, but it is not in accordance with my comment. There is still information out there that I feel is unnecessary. Therefore, I maintain my comment regarding abstract.
2. The authors added section 3.5 as a discussion of the research results. This chapter also aims to clear up my doubts regarding the conclusions. Unfortunately, I do not believe that this is a discussion in relation to the existing state of knowledge. Such a discussion should primarily focus on comparing the results obtained with what is currently known or used. What is required here is a reference to the current state of knowledge. Additionally, this chapter gives the impression of presenting conclusions. In the conclusion, the authors removed chnclusion 4 as I suggested. Overall, I think it would have been better without section 3.5, as it does not remove my comment about the discussion and conclusions.
As I wrote in the first review, the article does not raise any substantive objections to the presented research results, but I only believe that adapting the article in line with my comments would improve its scientific quality.
Therefore, I leave the final decision on the acceptance of the article to the Editors.
Author Response
Reply to Reviewer #4
General note: The Authors apologize for the insufficient quality of the previous revision. They have carefully reviewed the entire manuscript to satisfy the required corrections. The yellow color highlights all modifications in the text.
Comment: The authors provided a revised article and responded to my comments. The abstract has been corrected, but it is not in accordance with my comment. There is still information out there that I feel is unnecessary. Therefore, I maintain my comment regarding the abstract.
Answer: The Authors understand this point but respectfully disagree with the eminent Reviewer regarding the Abstract structure. In particular, the Authors believe that the description of the essential novelty, i.e., the development of the ultra-high-performance protection structure that requires bar reinforcement, is mandatory for understanding the necessity of this work. Since the typical refractory solutions do not require structural reinforcement. Therefore, the Authors shortened the Abstract again but preserved its description structure for the above reason.
Comment: The authors added section 3.5 to discuss the research results. This chapter also aims to clear up my doubts regarding the conclusions. Unfortunately, I do not believe that this is a discussion in relation to the existing state of knowledge. Such a discussion should primarily compare the results obtained with what is currently known or used. What is required here is a reference to the current state of knowledge. Additionally, this chapter gives the impression of presenting conclusions. In the conclusion, the authors removed conclusion 4, as I suggested. Overall, I think it would have been better without section 3.5, as it does not remove my comment about the discussion and conclusions.
Answer: The Authors appreciate this comment. Again, they want to point out the essential contribution of this research to engineering practice as the development of the ultra-high-performance protection structure that requires bar reinforcement, which involves information about the bond performance (investigated in this study). Therefore, they rewrote the first paragraph of Section 3.5 to clarify this essential aspect.
Comment: As I wrote in the first review, the article does not raise any substantive objections to the presented research results, but I only believe that adapting the article in line with my comments would improve its scientific quality. Therefore, I leave the final decision on the acceptance of the article to the Editors.
Answer: The Authors hope the above clarifications and corrections in the text made the manuscript acceptable for publication.